# On the Benefits of Early Fusion in Multimodal Representation Learning

**George Barnum,**
**Sabera Talukder**
**& Yisong Yue**
Department of Computation and Neural Systems, Neurobiology, Computing and Mathematical Sciences
California Institute of Technology
Pasadena, CA 91125, USA
{gbarnum,sabera,yyue}@caltech.edu

## Abstract

Intelligently reasoning about the world often requires integrating data from multiple modalities, as any individual modality may contain unreliable or incomplete information. Prior work in multimodal learning fuses input modalities only after significant independent processing. On the other hand, the brain performs multimodal processing almost immediately. This divide between conventional multimodal learning and neuroscience suggests that a detailed study of early multimodal fusion could improve artificial multimodal representations. To facilitate the study of early multimodal fusion, we create a convolutional LSTM network architecture that simultaneously processes both audio and visual inputs, and allows us to select the layer at which audio and visual information combines. Our results demonstrate that immediate fusion of audio and visual inputs in the initial C-LSTM layer results in higher performing networks that are more robust to the addition of white noise in both audio and visual inputs. Find the full paper at: https://arxiv.org/pdf/2011.07191.pdf

## 1 Introduction

Multimodal learning is important for many tasks, including audio visual speech recognition (Yu et al., 2020; Zhou et al., 2019; Su et al., 2017), emotion recognition (Park et al., 2020; Cao et al., 2014), multimedia event detection (Song et al., 2019), depth-based object detection (Wang et al., 2015b,a), urban dynamics modeling (Zhang et al., 2017), image-sentence matching (Liu et al., 2019), and biometric recognition (Song et al., 2019). In many cases, an individual modality does not contain sufficient information to classify the scene. Therefore, utilizing multiple modalities is crucial, particularly in complex tasks or domains prone to noisy data.

One important design decision in multimodal learning is how to best combine, or fuse, the different input modalities (Baltrušaitis et al., 2018; Li et al., 2018). Prior work on multimodal learning has largely relied on extensive unimodal featurization and other preprocessing before fusing the different modalities (Katsaggelos et al., 2015; Atrey et al., 2010). On the other hand, it is known that biological neural networks engage in multimodal fusion in the very early layers of sensory processing pathways (Schroeder & Foxe, 2005; Budinger et al., 2006). This divide between conventional multimodal learning and neuroscience suggests that a detailed study of early multimodal fusion could yield insights for improving multimodal representation learning.

In this paper, we study the benefits of early fusion in multimodal learning. To facilitate this study, we design a convolutional LSTM (C-LSTM) architecture that enables audio and visual input fusion

---

*Equal contributors.

2nd Workshop on Shared Visual Representations in Human and Machine Intelligence (SVRHM), NeurIPS 2020.

at various layers in the architecture. We find that early fusion outperforms late fusion, and that early fusion enables robust performance over a range of signal to noise ratios. These results shed new light on the power of immediate fusion as a means to improve model performance in the presence of noise. Integrating multimodal inputs as soon as possible can be generalized to other multimodal domains, such as audio-visual speech recognition and emotion recognition to increase their performance and representational power.

## 2 Related Work

**Multimodal Representation Learning.** Baltrušaitis et al. (2018) thoughtfully broke down the main problems within multimodal machine learning into five categories: representation, translation, alignment, fusion, and co-learning. Turk (2014) also posits that multimodal integration, also referred to as the fusion engine, is the key technical challenge for multimodal systems.

When dealing with fusion, there are currently two common paradigms: early fusion (Atrey et al., 2010) and immediate fusion (Katsaggelos et al., 2015). In early fusion, audio and visual modalities are first featurized before being passed to two independent modeling process units that do not differentiate between features from different modalities (Katsaggelos et al., 2015). On the other hand, immediate fusion is when the audio and visual modalities are first featurized and then sent to a joint modeling process unit (Katsaggelos et al., 2015). This unfortunate terminology does not take into account the possibility of fusing the inputs before any substantial featurization, which does occur in biological neural networks. Therefore, we will propose and utilize our own terminology below.

**Connections to Neuroscience.** In the brain, multisensory integration was traditionally believed to occur only after single modality inputs underwent extensive processing in unisensory regions (Schroeder & Foxe, 2005). However, we now know that in many species, including humans, multisensory convergence occurs much earlier in low level coritical structures (Schroeder & Foxe, 2005). In fact, primary sensory cortices may not be unimodal at all (Budinger et al., 2006). This may in part be because of individual neuron's abilities to be modulated by multiple modalities (Meredith & Allman, 2009). In a striking discovery, Allman & Meredith (2007) found that 16% of visual neurons in the posterolateral lateral suprasylvian that were previously believed to be only visually responsive were significantly facilitated by auditory stimuli. This philosophical departure from individual modality processing towards early multimodal convergence in neuroscience lays a promising groundwork for high-impact explorations in multimodal machine learning.

## 3 Models

In order to maintain the advantages of modality specific inductive biases, such as convolution or recurrence, while also allowing for the immediate fusion of audio and visual inputs, we created a multimodal convolutional long short term memory network that generates fused audio-visual representations with appropriate inductive biases. Our convolutional long short term memory, or C-LSTM, architecture performs the standard LSTM operations at every point of a convolution, using shared weights and separate cell and hidden state values. Further architectural details can be found in A.1 and A.2

Using our C-LSTM architecture, we constructed multiple different models in order to study the benefits of multimodal fusion.

- The full C-LSTM model that allows for fusion in the early layers (akin to how biological networks fuse sensory processing in early layers).
- Restricting fusion to the intermediate convolutional layers. Which in this case is fusion in the second layer.
- Restricting fusion to the fully connected layers (akin to prior work that performed unimodal featurization prior to fusion).
- Only processing visual or audio input.

We confirmed the ability of our model to effectively operate on both audio and visual data alone by occluding each of the modalities in 5.3.

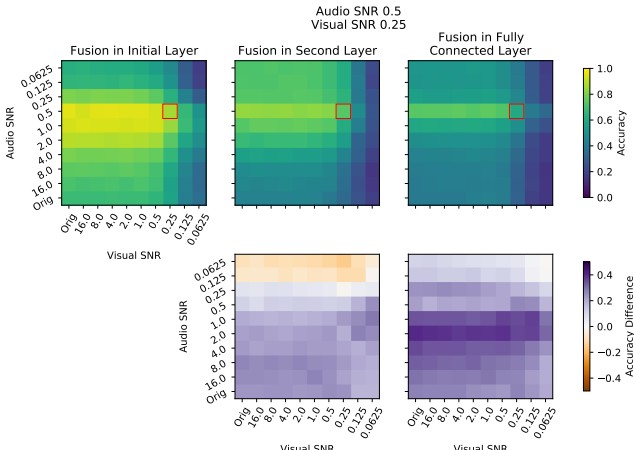

Figure 1: Comparing the performance of initial layer fusion, second layer fusion, and FC fusion models. The first row shows raw test accuracy at various signal to noise ratios. The second row shows the difference in accuracy between the late fusion models and the immediate fusion model at corresponding signal to noise levels. All models were trained with an audio SNR of 0.5 and a visual SNR of 0.25, labeled with a red box. Orig signifies the original audio or visual input.

# 4 Dataset

We constructed a multimodal dataset based on the well-known MNIST dataset (LeCun et al., 2010) and the Free Spoken Digit dataset (Jackson et al., 2018). We selected these datasets because of their tractability. This allows us to combine them to create a multimodal task which we know will be solvable. This allows us to artificially manipulate the difficulty of the task via the addition of noise. We prefer a regime in which we can break the system to understand its principles of fusion and representation. For more details see A.3.

# 5 Experiments

## 5.1 Fusion Comparison

To examine the value of early compared to late multimodal fusion, we created models which replace early multimodal C-LSTM layers with separate streams for each modality, as described in A.2. The initial layer fusion model, the second layer fusion model, and the fully connected layer fusion model were trained with the same signal to noise ratios of training data. We then tested the accuracy of each of these models for a range of values of the signal to noise ratios of both the audio and visual inputs. Then we compare the accuracy of the initial layer fusion model to the accuracy of each of the late fusion models.

As seen in Figure 1, the initial layer fusion model is more accurate not only for the signal to noise ratio that the models were trained at, but also for the majority of the other signal to noise ratios. In particular, the initial layer fusion model always outperforms or equally performs to the fully connected fusion model. Furthermore, the initial layer fusion model outperforms the second layer fusion model, except for when the audio input is degraded well beyond the audio training signal to noise ratio.

Additionally, initial layer fusion appears to allow the model to be much more robust to increases in the signal to noise ratio beyond the training values, especially in the case of increased audio SNR. The main characteristic of the SNR values in which initial fusion does not outperform late fusion is low audio SNR and relatively high visual SNR, and this only occurs in the case where fusion occurs in the second layer. This suggests that initial layer multimodal fusion encourages the multimodal model to use both input modalities.

While the specific areas and degree to which early fusion outperforms delayed fusion varies with the SNRs of the training data, the general trends are similar, and these fusion plots are representative of

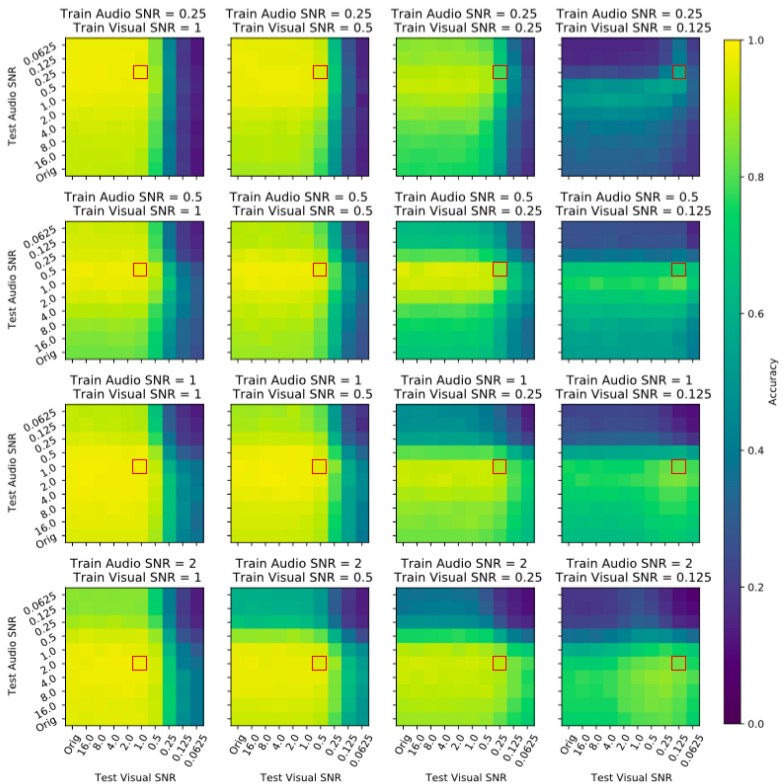

Figure 2: Accuracy of multimodal models trained on data with a range of signal to noise ratios, each for a range of signal to noise ratios of the testing data. Training SNRs labeled by red boxes.

models trained at other audio and visual SNRs; see A.4 for the same plot at other audio and visual training levels.

## 5.2 Robustness to Noise

An advantage of multimodal processing is the network's resilience to noise in the inputs. To examine our multimodal model's resilience to white noise we trained 16 models at distinct audio-visual SNR combinations and then tested each model in 100 different audio-visual SNR regimes. The results of these experiments can be seen in figure 2. In the bottom right quadrant of figure 2 we see that the model is mostly audio dependent. However, when the visual SNR is 1 and 0.5, there is strong visual dependence.

The addition of noise to both the training and test data is designed to provide a setting that allows the exploration of the limits of this model while using a simple dataset. These results mirror our expectations of how a multimodal model would behave both to various training and testing SNRs.

## 5.3 Comparison to Unimodal Models

To verify that joint audio visual representations are a result of both modalities, we tested our multimodal model on unimodal inputs by setting one input to zero. This created unimodal visual models and unimodal audio models without changing the underlying architecture. For each of these unimodal models, we trained the model at four SNR values, and tested the accuracy across our previously selected set of signal to noise ratios. The accuracy of these unimodal models for the SNR values is displayed in figures 3 and 4.

As expected, the unimodal models perform relatively well at the SNR that they are trained at or higher, although as the training SNR decreases, the models perform worse overall, and the perfor-

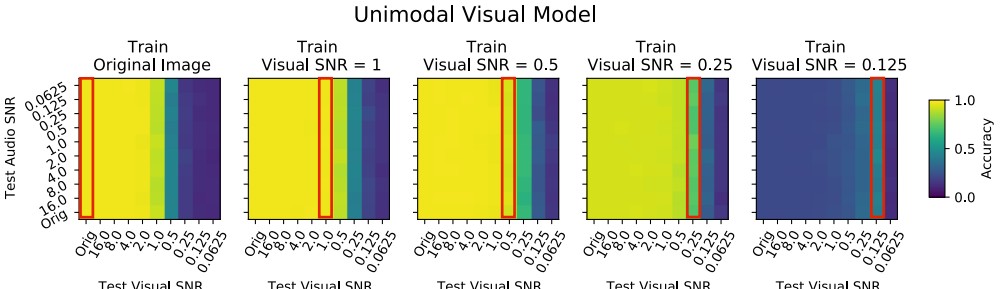

Figure 3: Performance of the C-LSTM model with only visual input trained on the original data as well as at various visual signal to noise ratios.

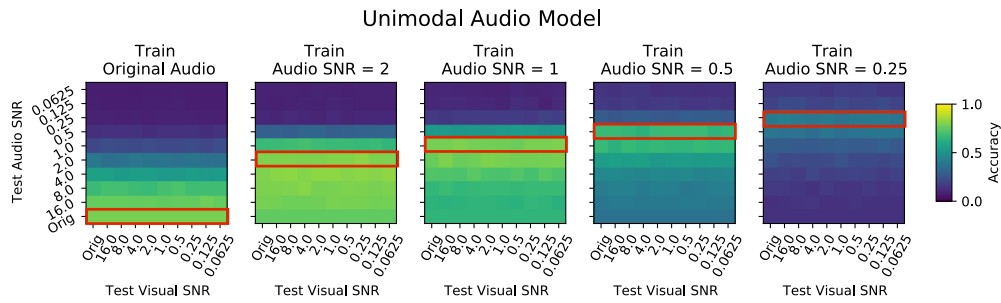

Figure 4: Performance of the C-LSTM model with only audio input trained on the original data as well as at various audio signal to noise ratios.

mance on higher SNR data decreases. This effect is particularly noticeable in the audio only model, figure 4.

Additionally, these unimodal models demonstrate that the C-LSTM architecture is at minimum capable of using information from either modality to perform the classification task. However, the difference in performance between the audio and visual unimodal models shows that the model is not equally sensitive to each modality or to noise in each modality. This discrepancy motivated us to choose an audio SNR of 0.5 and a visual SNR of 0.25 for the model investigation discussed above in figure 1. Choosing these audio and visual SNR values allowed us to investigate early fusion in a setting where multimodal fusion would be beneficial, as discussed above in section 3.

## 6 Conclusion

In this paper, we developed a C-LSTM architecture to investigate the effects of fusion depth on noise robustness. Motivated by the neuroscientific literature suggesting that sensory inputs are combined early in processing, we proposed that truly immediate fusion of modalities would provide benefits in neural networks as well. Our initial layer fusion model demonstrates robustness to changes in input noise and an improvement in accuracy relative to late fusion models with analogous architectures. Future directions include investigating the effects of immediate multimodal fusion in deeper networks and on problems with more inherent difficulty, as well as the extension of immediate multimodal processing to other multimodal domains such as translation, alignment, and co-learning. However, we believe that the results demonstrated here and in A.5 show the importance of truly immediate fusion and could help with other domain specific multimodal learning tasks.

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

# A   Appendix

## A.1   Multimodal Convolutional LSTM Model

At each point of convolution, the first layer takes:

- The section of the input image to be multiplied by our convolutional kernel, denoted as $\mathbf{v}$.
- The section of the hidden state to be multiplied by our convolutional kernel, denoted as $\mathbf{h}_{t-1}$. It is initialized to zeros for the first time step in the first layer.
- The spectrogram value of the audio input, at a given time step, denoted as $\mathbf{a}_t$.

Our model then computes the LSTM gate values using: $\mathbf{v}$, $\mathbf{h}_{t-1}$, and $\mathbf{a}_t$ via the equations in Figure 5, under the *Initial LSTM Gate Values* section. The initial C-LSTM layer produces a single multimodal tensor that combines information from the audio and visual inputs. As in the standard LSTM architecture, the hidden state, $\mathbf{h}_t$, of the previous layer is used as the input, $\mathbf{x}_t$, of the current layer. Therefore, in our subsequent LSTM layers the gate values at each location are computed from the section of the combined multimodal input, $\mathbf{x}_t$, to be multiplied by our convolutional kernel. Equations found in Figure 5, under the *Subsequent LSTM Gate Values* section. By applying the LSTM operations at each location of a convolution, this architecture allows the LSTM cells to respond to the spatial information from the visual domain as well as the temporal information of the audio domain. This architecture enables us to study the mixing of signals at the initial, second, and fully connected layer while maintaining the same inductive biases that are beneficial in processing image and sequence data.

**Varying the Fusion Level.** The full C-LSTM approach described above performs early fusion, i.e., mixing the modalities starting in the very first layer. However, our framework can be easily modified to only allow for fusion in later layers, or to mask out one modality all together.

## A.2   Training & Modeling Details

For the full model, we used: an initial merge layer (64 units and $3 \times 3$ kernels, and a $2 \times 2$ max pool layer), a second multimodal C-LSTM layer (64 units and $3 \times 3$ kernels, and a $2 \times 2$ max pool layer), a dense layer (128 units and ReLU activation), and then a final dense output layer (10 units).

The second layer fusion model consists of a separate convolutional layer with 64 units and $3 \times 3$ kernels and an LSTM layer with 64 units. These layers feed into a C-LSTM layer, with 64 units and

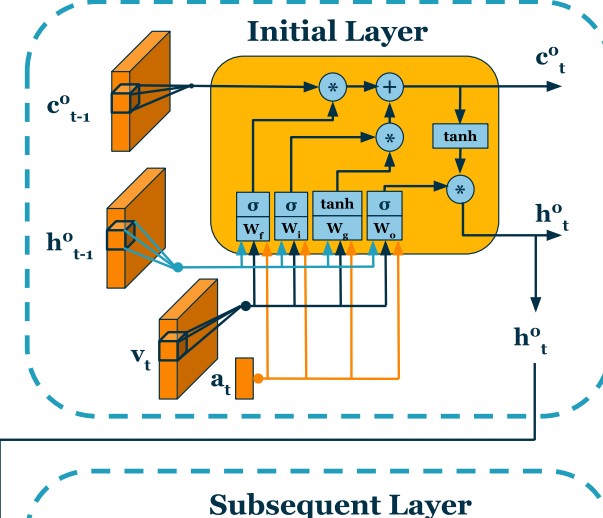

**Initial LSTM Gate Values:**

$$f_t = \sigma(W_f[\mathbf{h}_{t-1}, \mathbf{x}_t] + b_f)$$
$$i_t = \sigma(W_i[\mathbf{h}_{t-1}, \mathbf{x}_t] + b_i)$$
$$g_t = \tanh(W_g[\mathbf{h}_{t-1}, \mathbf{x}_t] + b_g)$$
$$o_t = \sigma(W_o[\mathbf{h}_{t-1}, \mathbf{x}_t] + b_o)$$

**Cell State Update:**

$$c_t = f_t c_{t-1} + i_t g_t$$

**Hidden State Update:**

$$h_t = o_t \tanh(c_t)$$

**Subsequent LSTM Gate Values:**

$$f_t = \sigma(W_f[\mathbf{h}_{t-1}, \mathbf{x}_t] + b_f)$$
$$i_t = \sigma(W_i[\mathbf{h}_{t-1}, \mathbf{x}_t] + b_i)$$
$$g_t = \tanh(W_g[\mathbf{h}_{t-1}, \mathbf{x}_t] + b_g)$$
$$o_t = \sigma(W_o[\mathbf{h}_{t-1}, \mathbf{x}_t] + b_o)$$

Figure 5: The first two layers of the multimodal convolutional long-short term memory network, and the equations used to compute the gate and update values. $f_t$ is the forget gate, $i_t$ is the input gate, $g_t$ is the cell gate, and $o_t$ is the output gate. $W$'s are the corresponding weight matrices, and $b$'s the corresponding bias values. $\sigma()$ is the sigmoid function. $\tanh()$ is the hyperbolic tangent function.

$3 \times 3$ kernels, a $2 \times 2$ max pool layer, a dense layer with 128 units and ReLU activation, and finally a final dense output layer with 10 units.

The fully connected layer fusion model consists of separate processing streams for the visual and audio data. The visual stream consists of two convolutional layers with 64 units and $3 \times 3$ kernels, while the audio stream consisting of two LSTM layers with 64 units. The output of the convolutional layers and the last timestep of the output of the LSTM layers are concatenated and fed into a dense layer with 128 units and ReLU activation, then a final dense output layer with 10 units.

We trained all models using the Adam optimizer (Kingma & Ba, 2014) with a learning rate of 0.001. Each model is trained on 87516 examples randomly selected from our multimodal dataset (which in total contains 11202076 training data points, 174490 validation data points, 175389 test data points). The models can be trained on any combination of audio and visual SNRs.

### A.3   Input data

For each of these datasets, we first combined all the data, testing and training, into one dataset, then split each of the two datasets into training, testing, and validation sets with an $8 : 1 : 1$ ratio. Next, for each of the training, testing and validation splits, we created a dataset containing all image and audio pairs with the same label. We then augmented the data by adding white noise to the image and audio data such that the signal to noise ratio could be chosen, as shown in Figure 6. This allows us to explore our model's response to degradation in either the image or audio input, while utilizing noise techniques commonly applied to inputs (Borji & Lin, 2020). We then front padded each audio input

with zeros such that all the audio examples are of equal length. Finally, we took the spectrogram of each audio input, with 400 samples, 201 frequency bins, and a stride of 200 samples.

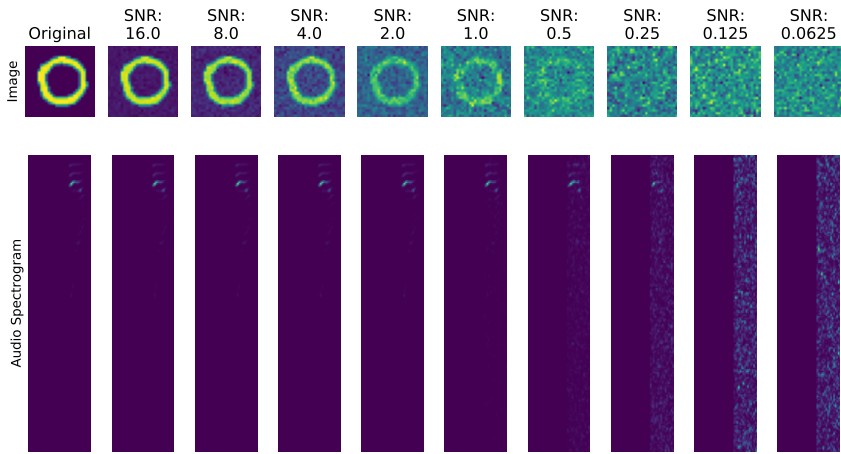

Figure 6: The same input example where white noise has been applied at various SNR values, with the full audio spectrogram.

## A.4 Late Fusion

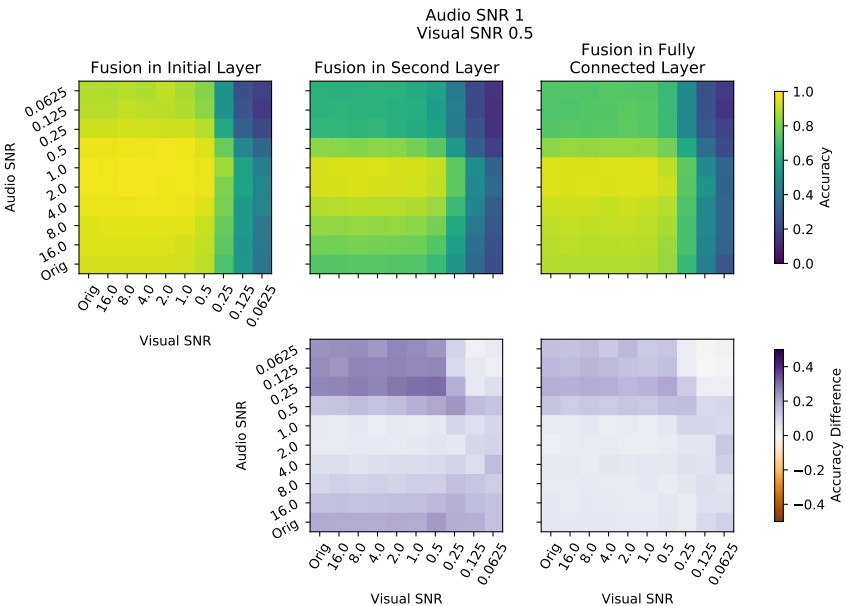

Figure 7: Late fusion model where audio SNR = 1, and visual SNR = 0.5. Fusion in the initial layer outperforms the fusion in the fully connected layer.

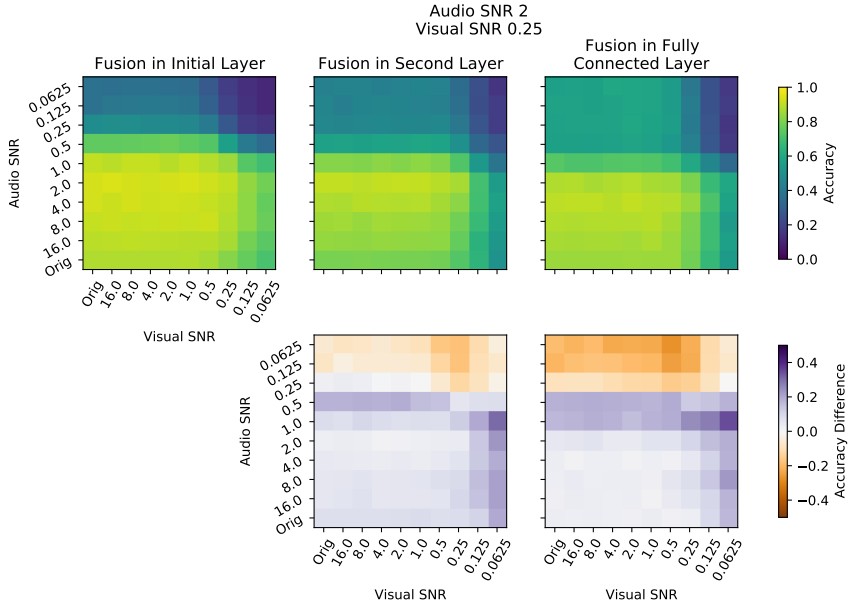

Figure 8: Late fusion model where audio SNR = 2, and visual SNR = 0.25. Fusion in the initial layer outperforms the fusion in the fully connected layer for audio SNR values greater than 1.

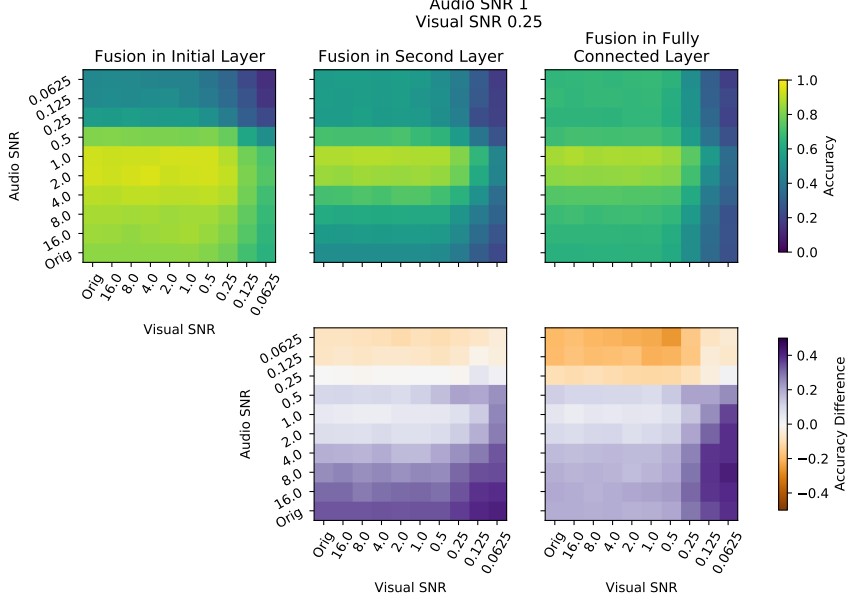

Figure 9: Late fusion model where audio SNR = 1.0, and visual SNR = 0.25. Fusion in the initial layer outperforms the fusion in the fully connected layer for audio SNR values greater than 1.

## A.5   Model Inspection

In order to examine the contribution of audio and visual inputs to the performance of our multimodal classifier, we considered the state of the network at intermediate timesteps in the recurrent processing of the audio inputs. In figure 11 we display the activations of the final layer of the network across timesteps for a single representative example at various signal to noise ratios. These final layer activations correspond to the classification of the input into each of the ten digit classes. We display the activations in response to four SNR scenarios: the original input, a scenario where the audio

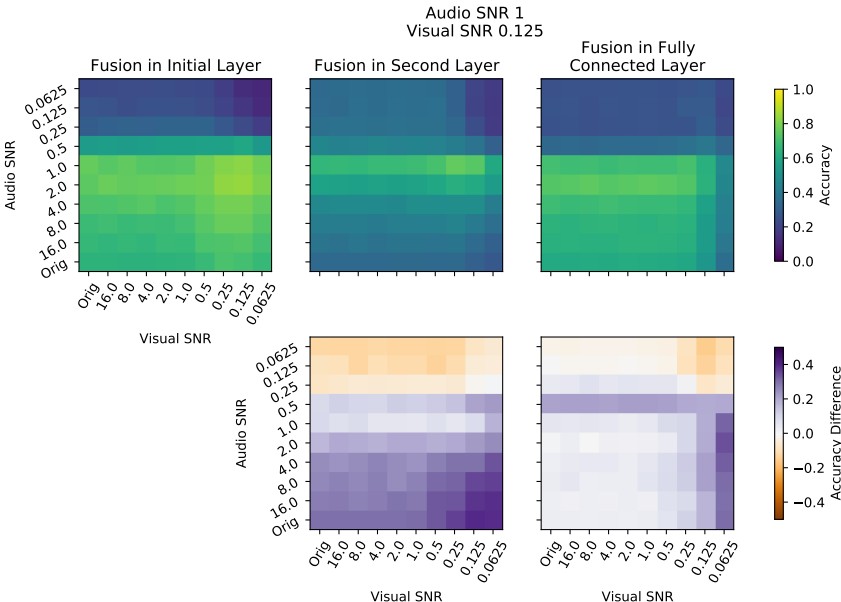

Figure 10: Late fusion model where audio SNR = 1, and visual SNR = 0.125. Fusion in the initial layer outperforms the fusion in the fully connected layer for audio SNR values greater than 1.

input has a higher SNR than the visual input, a scenario where the visual input has a higher SNR than the audio input, and a scenario where the visual input and the audio input have equal SNR.

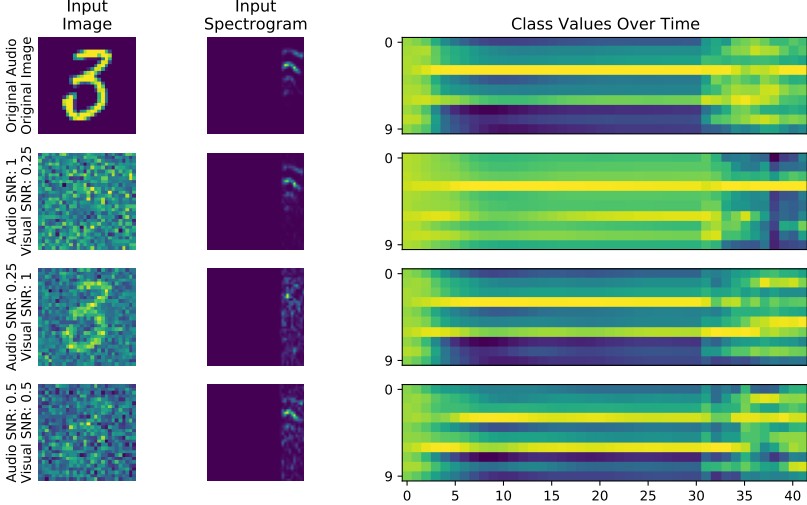

Figure 11: The values of the final layer of the multimodal layer across timesteps of the C-LSTM for a representative example at various signal to noise ratios.

This visualization demonstrates the value of the multimodal input in this network. Because the image information is available to the network for the entire length of the audio, the network initially responds to the image without audio input, because the audio is front zero-padded. As the network is evaluated along the time dimension of the audio input, information from the audio input becomes available to the network. Therefore, comparing the activations of the final layer can illuminate the contribution of each modality to the network.

In this example, the original input demonstrates the capability of the same model to correctly classify the digit in both the visual and audio regime, because both before and during the audio input, the correct class, 3, is assigned the maximum value. In the example where the audio input has an SNR of 1.0 and the visual input has an SNR of 0.25, as well as the example with audio SNR of 0.5 and visual SNR of 0.5, the contribution of the audio input becomes more evident, as in both of these scenarios, before the audio input is available the model shows more confusion, both with all the other classes, and in particular with class 6. In these examples, as the audio information becomes available to the network, the correct class comes to dominate the activations.

Additional examples of final layer activations.

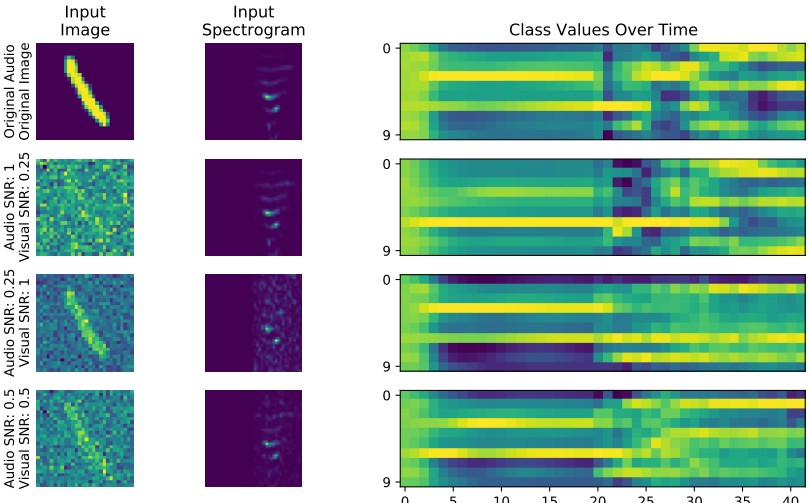

Figure 12: Additional example of values of the final layer of the multimodal layer across timesteps of the C-LSTM for a representative example at various signal to noise ratios.

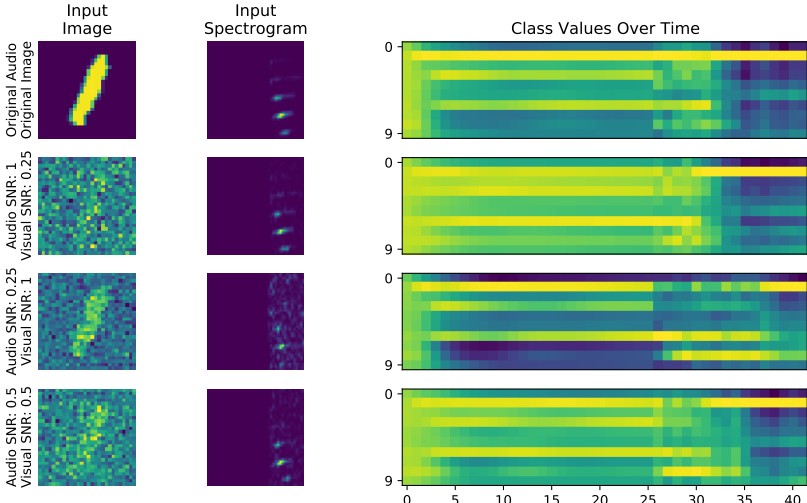

Figure 13: Additional example of values of the final layer of the multimodal layer across timesteps of the C-LSTM for a representative example at various signal to noise ratios.

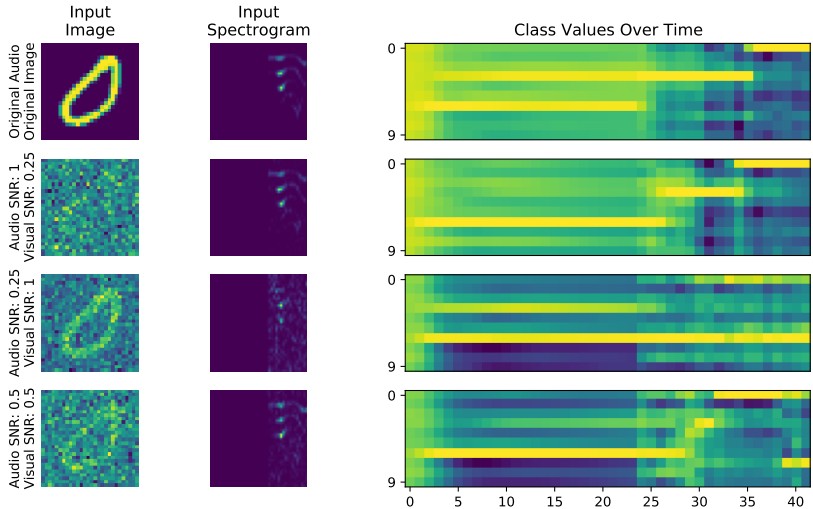

Figure 14: Additional example of values of the final layer of the multimodal layer across timesteps of the C-LSTM for a representative example at various signal to noise ratios.

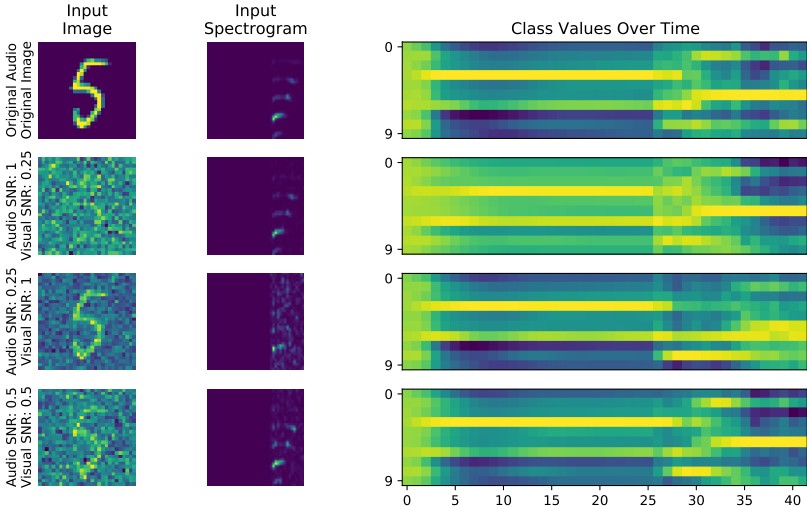

Figure 15: Additional example of values of the final layer of the multimodal layer across timesteps of the C-LSTM for a representative example at various signal to noise ratios.

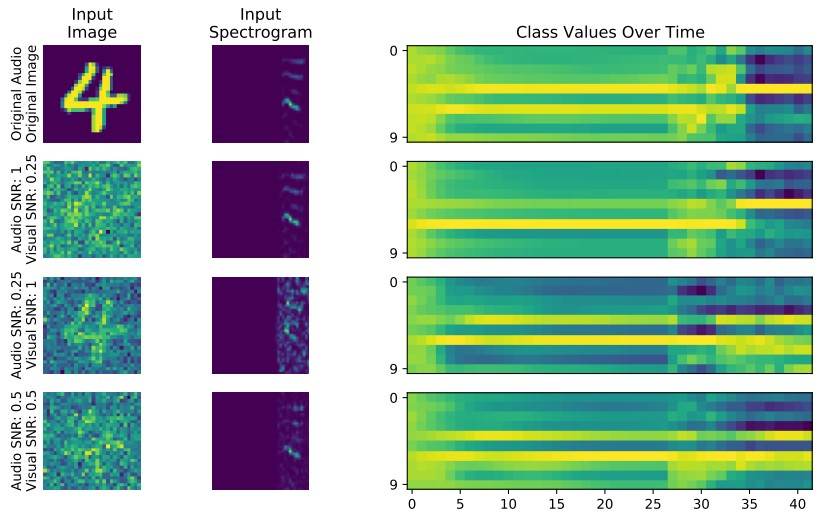

Figure 16: Additional example of values of the final layer of the multimodal layer across timesteps of the C-LSTM for a representative example at various signal to noise ratios.

