# OpenReview forum: "On the Benefits of Early Fusion in Multimodal Representation Learning"
_NeurIPS.cc/2020/Workshop/SVRHM — SVRHM@NeurIPS Poster_

### Official Review · AnonReviewer2 · 2020-10-27
**Nice computational support for early fusion**

**Rating:** 7
**Confidence:** 4

**Review:**

This paper investigates at what stage of processing it is best to fuse multimodal input. It therefore generates a toy dataset combining written (MNIST) and spoken (Free Spoken Digit) data to mimic multimodal input.
It then compares different levels in artificial neural networks to merge the input and measures the resulting performance. Since the task without noise would be too easy they add noise to the datasets to make the task harder.

My first concern would be the data at hand. The visual and audio stimulus don't actually have any relation besides the class label. That leads to an unnatural setting and is even as a toy dataset not really reflecting the actual challenge. Also adding noise to the images renders it a multimodal denoising task which might not be the same as a setting where both image and audio are naturally hard (and not via noise).

A second concern would be with regards to capacity. I would like to ruled capacity issues out for this analysis. It would be interesting to see an analysis with a deeper and more overparametrized network to see at what stage it is optimal to merge the input signals.

Overall I think the paper provides careful evaluation and is not overstating its results and should therefore be accepted.

47: join -> joint (typo?)

---

> ### Public Comment · ~George_Barnum1 · 2020-12-07
> **Response Reviewer 2**
>
> Reviewer 2, thank you for the time you took to review our paper; we really appreciate it! We are pleased you thought that we performed a careful evaluation while not overstating our results.
>
> Addressing Concerns:
>
> While it is true that the artificial dataset does not fully reflect a real world situation where both modalities are produced by related processes, the fact that we demonstrate gains from early fusion relative to late fusion when the modalities are disconnected leads us to believe that this principle would generalize to situations where the modalities are more related.
>
> We agree that the comparison between a multimodal denoising task and a multimodal task where the inputs are naturally hard is not one to one. However, we think that the general behavior of early fusion should generalize between this artificial task and natural ones. In future work we hope to explore these questions on a more naturalistic multimodal task.
>
> We agree that testing the phenomenon we studied on deeper networks would be beneficial, and will incorporate this into future iterations of the paper.

---

### Official Review · AnonReviewer3 · 2020-10-28
**Interesting problem and motivation, however experiments have limited scope**

**Rating:** 5
**Confidence:** 4

**Review:**

The paper focuses on the demonstration that early fusion of data is preferable compared to later staged fusion.
The study focuses on an interesting aspect of sensor fusion, whether it should be early in processing or later.

There seem to be a few major limitations with the experiment construction of the work. If the goal is to compare early versus late fusion, a starting point point could be to concatenate input representations together before a network, and then for other runs, fuse at higher/later stages of the network after allowing some pre processing. It is unclear to this reviewer why the authors chose to adopt a more complex strategy using C-LSTMs. Further, given that the more simpler experiments have not been explored, the experiments seem limited in scope and applicability to more common network architectures.

That said, the paper offers results for robustness of noise and on a simple easy to investigate dataset. This reviewer encourages the authors to continue this direction of work however consider more simpler hypothesis to test first and perhaps a few more challenging datasets to test the limits of their approach.

---

> ### Public Comment · ~Sabera_Talukder1 · 2020-12-07
> **Response Reviewer 3**
>
> Reviewer 3, thank you for taking the time to review our paper!
>
> The immediate concatenation of the input representations, while possible, would sacrifice the ability to take advantage of the structure of the input given the differences in structure between time series and image data. Since many neural network architectures for audio and image data rely heavily on the structure of the data, we decided that such an approach would be less relevant to potential applications.
>
> On your point about fusing at later stages, The C-LSTM model reduces to the experiments that you are suggesting. We apologize if we didn’t clearly explain all of the ablation experiments performed. The reason we created the more complicated C-LSTM framework was so that we could study ablations in a more comparable manner.

---

### Official Review · AnonReviewer1 · 2020-10-30
**A pleasant read with neat experiments and promising results**

**Rating:** 9
**Confidence:** 4

**Review:**

## Summary of the paper

This article proposes immediate fusion of data modalities for multimodal representation learning. The authors argue that the brain performs multimodal integration very early in the sensory processing to motivate a similar solution for artificial systems. To explore this idea, the paper presents the results of training a bimodal architecture, a convolutional LSTM, on a bimodal data set that combines MNIST with Free Spoken Digits. From their results, the authors conclude that immediate multimodal fusion provides better performance and robustness than late integration.

## Summary of merits and concerns

### Merits

+ The topic is relevant and is likely to receive increased attention in the near future, especially if more challenging multimodal tasks can be appropriately designed for experimentation.
+ The paper is very easy to read, well-written, the results are presented very neatly in the figures.
+ Besides the results, the paper introduces an interesting network architecture that combines visual and audio inputs flexibly, as well as a simple multimodal data set, that may be used in future research.

### Concerns

- Since the empirical results are obtained on a newly constructed data set, not very challenging and carefully controlled, the results have intrinsically limited impact.
- For some of the contributions to be useful for the community, the code and data should be made available. Will they be?
- After reading the paper, there a few things for which I need further clarification, which I detail below.

## Evaluation and justification

I assess this submission very positively. While it is not the most important aspect to take into account, the presentation is great: the paper is easy to read, the introduction nicely motivates the problem, the review of related work is brief but relevant (perhaps more recent papers on multimodal representation learning should be reviewed), the experimental setup is fairly clear and the figures showing the results are quite neat. Beyond this, the topic is relevant for both the machine learning community and the neuroscience community, and the paper draws relevant connections between the two worlds. Furthermore, although of limited impact due to the nature of the data and the tasks, the results and conclusions are promising. I believe that this paper can be of interest and inspiring for the SVRHM audience and others and therefore I strongly recommend the acceptance for the workshop.

## Questions

I would like to have a couple of questions clarified:

- In Figure 1 and others, why is the performance worse with higher signal to noise ratio? This seems counter-intuitive.
- Why are not results of training with the original SNR presented in the paper?

I may be missing something, but these observations make be think that there may be some issues with the data set and the task that generate some artifacts like these.

## Minor comments and potential typos identified

I would like to note below some minor comments, suggestions and potential typos that the authors may take into account for future versions of the paper:

- Typo: "However, we now know that in many species, including humans, _that_ multisensory convergence occurs much earlier": extra _that_
- Suggestion: propose and consistently use a terminology to refer to each model (immediate, second layer, fully connected, etc.)
- Suggestion: in the figures, mark graphically the audio and visual SNR used for training, for example with a vertical and horizontal line.
- The performance of the unimodal visual model is near to perfect, as it is well-known from the extensive literature on the MNIST data set. However, the performance is far from perfect on the Free Spoken Digits data set. It would be desirable to construct a model that would solve the audio task as well as the visual task for easier comparisons.

---

> ### Public Comment · ~Sabera_Talukder1 · 2020-12-07
> **Response Reviewer 1**
>
> Reviewer 1, thank you for the time you took to review our paper! We’re happy to see that you believe our study is relevant both to the machine learning and neuroscience communities!
>
> Addressing Concerns:
>
> We understand that working on a relatively simple, and newly constructed data set limits the ability to generalize our findings. However, we wanted to thoroughly explore and validate our hypothesis before trying many different data domains. We hope to perform additional experiments on visual + audio datasets to further validate our findings.
>
> We hope to make the code and data available, once we have had the full paper accepted at a conference.
>
> We believe the performance is worse at higher signal to noise ratios because the model is trained on data with a lower signal to noise ratio, and therefore performs worse when it encounters inputs with different statistics, even if the different statistics actually reflect a higher signal to noise ratio.
>
> In the supplemental figures 4 and 5 the performance of the unimodal versions of the model trained on the original images and audio are shown. Based on these results, we chose to examine the performance of the multimodal model in more difficult noise regimes, so that differences caused by multimodal fusion would not be masked by saturation of the model performance. In future iterations of the paper we will include the results of training the multimodal model on the original data for completeness.
>
> We agree that we should construct a model that solves the audio task as well as the visual task. We will run these experiments for future iterations of the paper.
>
> For your other minor comments, we have addressed the suggestions and updated the paper to reflect them.
>
> We updated the terminology to be initial layer, second layer, and fully connected layer fusion models.

---

### Decision · Program_Chairs · 2020-11-02

Accept (Poster)